# ARTS: ALLEVIATING HALLUCINATIONS IN LARGE VISION–LANGUAGE MODELS VIA REDUNDANCY-AWARE TOKEN SELECTION

## ABSTRACT

Large Vision–Language Models (LVLMs) demonstrate significant potential in multimodal tasks, yet they are prone to hallucinations, where generated outputs deviate from the visual evidence. A mainstream approach to mitigate hallucinations in LVLMs is to develop training-free decoding strategies. Most of these methods posit that hallucinations stem from **insufficient attention to relevant information** and therefore focus on strengthening the model's utilization of informative content. Beyond this perspective, we reveal a new source of hallucination: **Visual tokens in intermediate decoder layers often become redundant or noisy, thereby misleading multimodal reasoning.** Next, we evaluate commonly used token-importance metrics and observe that they cannot effectively identify redundant visual tokens in this context. To address this problem, we introduce **ARTS**, a decoding strategy that first reintegrates the original visual embeddings to enrich essential visual information, and then employs a novel sink-token-based method to select important visual tokens in intermediate decoder layers. Extensive experiments on multiple benchmarks and LVLM architectures demonstrate that our approach consistently reduces hallucinations and improves factual alignment.

## 1 INTRODUCTION

Large Vision-Language Models (LVLMs) have recently shown remarkable potential and widely used in various open-ended visual understanding tasks such as image captioning, visual question answering, and multimodal dialogue (Ye et al., 2023; Lee et al., 2024; Li et al., 2023a; Liu et al., 2023b; Zhu et al., 2023). However, hallucination is still a critical limitation of current LVLMs (Yin et al., 2024; Liu et al., 2023a) — generating text that appears fluent and coherent but includes incorrect or image-irrelevant content. Such hallucinations severely compromise reliability and trustworthiness, thereby constraining their adoption in real-world scenarios Chen et al. (2024b).

One mainstream strategy to address hallucinations in LVLMs is through training-free methods that operate solely at inference, such as contrastive decoding (Chuang et al., 2023; Wang et al., 2024c; Leng et al., 2024; Favero et al., 2024; Huo et al., 2024), which contrast outputs from different model states (e.g., original vs. perturbed inputs, or shallow vs. deep layers) to suppress spurious responses and enhance factual grounding. Another line of work (Wang et al., 2025; 2024b; Tang et al., 2025b) aggregates information from multiple decoder layers to maintain semantic consistency and mitigate information loss during the decoding process. A further approach (Huang et al., 2024b; Kang et al., 2025; Tang et al., 2025a) redistributes attention from "sink tokens" to more informative visual or textual content, improving the model's focus on relevant evidence. These approaches effectively mitigate hallucinations by enhancing the model's utilization of informative content. However, they operate under a common assumption that hallucinations primarily stem from insufficient attention to relevant information. This leads to a fundamental question: ***Beyond insufficient attention to relevant information, could hallucinations also arise from the model processing redundant or misleading information that conflicts with accurate visual evidence?***

To investigate this hypothesis, we conduct layer-wise token pruning experiments across different stages of LVLM processing, as shown in Figure 1b. Our findings reveal a surprising pattern: Ran-

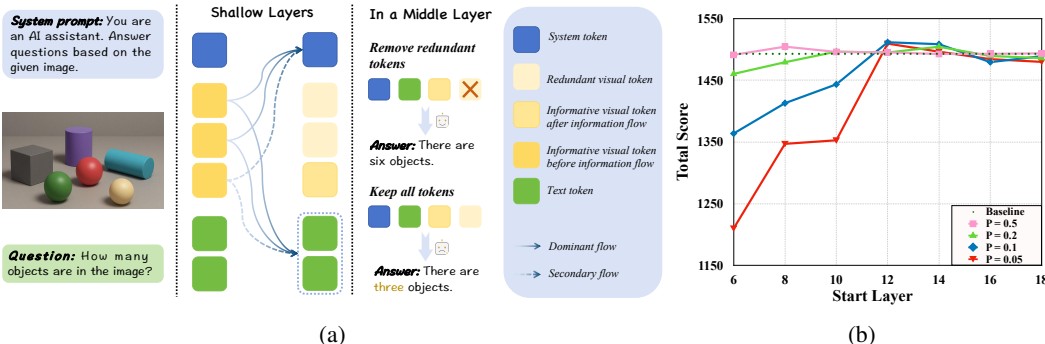

(a)                                                             (b)

Figure 1: (a) Illustration of visual information flow and redundancy in LVLMs. In shallow layers, visual information flows into system and text tokens. In the intermediate layers, most visual tokens become redundant while only a few still preserve meaningful visual signals. Pruning these tokens prevents redundant information from further propagating into textual representations, thereby reducing hallucinations. (b) Performance of LLaVA-1.5-7B on MME dataset Fu et al. (2024) under random visual token pruning with different retain ratios across layers. The subsequent layers exhibit a similar trend to layer 16, with scores remaining slightly below the baseline.

domly pruning visual tokens in intermediate layers (12-14) actually improves model performance. This uncovers a new source of hallucinations, arising from redundant or weakly aligned visual tokens in intermediate layers that disrupt multimodal reasoning. Recognizing this issue naturally raises a critical methodological question: ***How can we systematically distinguish between informative and redundant visual tokens in intermediate layers, where traditional importance metrics may no longer apply?*** We first evaluate two widely used token-importance metrics, originally designed for accelerating inference of LVLMs, in this context as shown in Section 3.2 and find their significant limitations: **1.Visual-CLS attention** (Zhang et al., 2024a;b; Wang et al., 2024a; Huang et al., 2024a) shows limited effectiveness because CLS tokens aggregate primarily visual information but lack the cross-modal context necessary for accurate importance assessment in multimodal scenarios. **2.Visual-Text attention** (Huo et al., 2024; Xing et al., 2024; Chen et al., 2025; Zhao et al., 2025; Cao et al., 2023; Zhang et al., 2024c) also proves to be insufficient because text tokens aggregate limited information in intermediate layers, making them poor indicators of the importance of visual tokens. These limitations underscore the necessity of developing new, principled criteria for identifying informative visual tokens in intermediate layers of LVLMs.

To overcome these limitations, we propose **ARTS** (Redundancy-**A**wa**R**e **T**oken **S**election) that leverages sink-visual attention, representing a novel application of sink tokens. Recent studies (Wang et al., 2023; Chen et al., 2024a; Huang et al., 2024b) have identified sink tokens as special positions that attract disproportionate attention and serve as global information aggregators, accumulating knowledge from across the entire sequence. We further validate this in Section 4. While prior works (Huang et al., 2024b; Kang et al., 2025; Tang et al., 2025a) primarily focused on mitigating the negative effects of sink tokens by reallocating attention away from them, we propose the first approach to harness sink tokens as anchors for importance calculation. Our key insight is that sink tokens, having aggregated rich cross-modal information by intermediate layers, provide superior queries for assessing visual token relevance. Additionally, we reintegrate original visual embeddings before selection to ensure that essential visual information is preserved, creating a comprehensive framework that both eliminates harmful redundancy and maintains critical visual signals. Extensive experiments on multiple benchmarks demonstrate that our method substantially reduces visual hallucinations and improves factual accuracy in diverse LVLMs without additional training. Our key contributions are as follows:

- We identify a new source of hallucinations in LVLMs: In intermediate layers, most visual tokens carry redundant information that misleads the model reasoning and induces hallucinations.

- We demonstrate that existing token-importance metrics cannot effectively identify informative visual tokens at intermediate layers.

- We propose a novel approach that leverages cross attention between visual tokens and sink tokens to identify redundancy while reintegrating the original visual information, thereby alleviating hallucinations.

## 2 RELATED WORK

**Large Vison Language Models** LVLMs have evolved from BERT-style multimodal encoders (Lu et al., 2019; Tan & Bansal, 2019) to decoder-based systems that feed visual tokens into pretrained LLMs for unified generation (Touvron et al., 2023; Chiang et al., 2023). End-to-end pretraining improved cross-modal alignment (Jia et al., 2021; Radford et al., 2021), and instruction-tuned systems such as LLaVA and InstructBLIP advanced open-ended tasks (Liu et al., 2023b; Dai et al., 2023). Subsequent work explored scaling and token-efficiency: LLaVA-1.5 redesigned the vision–language connector (Liu et al., 2024); BLIP-2, InstructBLIP, and MiniGPT-4 employed learnable query transformers to fuse modalities and reduce image tokens (Li et al., 2023a; Dai et al., 2023; Zhu et al., 2023); InternVL combined a stronger vision encoder, dynamic high-resolution input, and high-quality bilingual data (Chen et al., 2024c). In parallel, the mPLUG-Owl family adopts a modular design with modality collaboration, improving instruction following and multimodal reasoning (Ye et al., 2023; 2024b;a). In our experiments, we evaluate on InstructBLIP2, MiniGPT-4, LLaVA-1.5, and mPLUG-Owl.

**Mitigating Hallucinations in LVLMs** Recent research has investigated training-free approaches to hallucination mitigation, with a focus on generation-time interventions at inference. DoLa (Chuang et al., 2023) contrasts logits from shallow and deep layers to surface factual knowledge, while VCD (Leng et al., 2024) extends this idea to LVLMs by comparing outputs from original and perturbed visual inputs. Other inference-time strategies refine consistency across layers: DAMO (Wang et al., 2025) enforces cross-layer agreement with momentum updates, DCLA (Tang et al., 2025b) aggregates hidden states as semantic references. Additional works mitigate hallucination by reducing reliance on sink tokens (Huang et al., 2024b; Kang et al., 2025; Tang et al., 2025a).

**Token Pruning in LVLMs** Token reduction has been widely studied as a way to reduce the computational burden of LVLMs by discarding redundant visual tokens. FastV (Chen et al., 2024a) first pointed out this redundancy and removed tokens with low attention scores after the early layers of the LLM. Among existing methods, one line of work computes attention between visual tokens and text tokens, using the resulting weights as importance scores to progressively prune less relevant tokens (Huo et al., 2024; Xing et al., 2024; Chen et al., 2025; Zhao et al., 2025; Cao et al., 2023; Zhang et al., 2024c). Another line instead measures the attention between visual tokens and a global "[CLS] token" that summarizes the visual input, ranking and discarding tokens with the lowest relevance (Zhang et al., 2024a;b; Wang et al., 2024a; Huang et al., 2024a). While these approaches achieve substantial acceleration by reducing the number of tokens processed, they frequently incur a degradation in performance.

## 3 EMPIRICAL PRIMER

### 3.1 VISUAL REDUNDANCY HYPOTHESIS

We begin by examining the effect of randomly pruning visual tokens at different decoder layers, as shown in Figure 1b and find that only pruning visual tokens at mid layers (12-14) consistently improves model performance even at a relatively small retain ratio of 0.05, while similar pruning in shallow layers degrades performance and deeper layer pruning has minimal effect. This is because, during shallow layer processing, visual information progressively flows from visual tokens into textual representations, as established by (Chen et al., 2024a; Yin et al., 2025). Toward the end of this information transfer, many intermediate-layer visual tokens retain redundant or incomplete representations that no longer align with the integrated multimodal context, as shown in Figure 1a. Pruning them at this stage prevents such redundant information from further propagating into textual representations and allows the remaining informative visual evidence to interact more coherently with textual representations, which in turn mitigates hallucinations. Hence, a natural question we ask: **Are there more effective strategies for pruning visual tokens beyond random pruning?**

## 3.2 RE-EXAMINING EXISTING PRUNING METHODS AT MID LAYERS

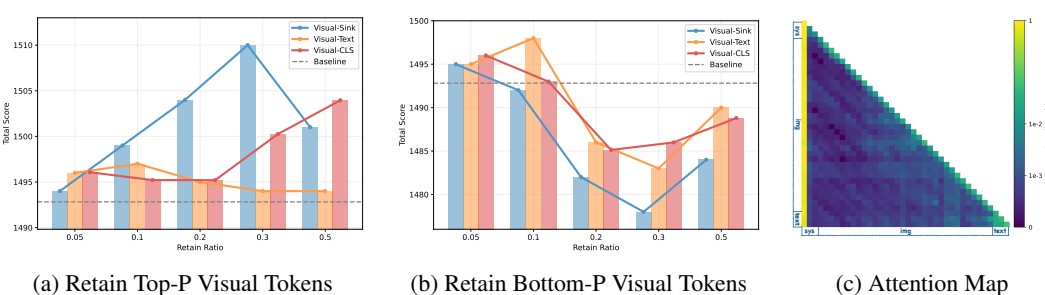

| (a) Retain Top-P Visual Tokens | (b) Retain Bottom-P Visual Tokens | (c) Attention Map |

Figure 2: (a) (b): Comparison of three visual token selection methods by retaining Top-P visual tokens 2a and Bottom-P visual tokens 2b at the 13th decoder layer of LLaVA-1.5-7B on Perception Subset of MME (Fu et al., 2024) dataset. (c) Attention maps at the 13th decoder layer of LLaVA-1.5-7B.

To answer the question in Section 3.1, we first evaluate two widely used token-importance metrics in our context: (i) visual–text attention, which is typically applied in shallow decoder layers, and (ii) visual–CLS attention, which is commonly used within the image encoder (details in Appendix B). Since our findings reveal that visual token redundancy emerges primarily in the middle layers, we re-examine the effectiveness of these metrics at intermediate depths. We retain visual tokens with the highest and lowest scores computed by the two methods under varying retain ratios. As shown in Figure 2a and 2b, visual-text attention is almost incapable of identifying informative tokens, as evidenced by the fact that when retaining top-p tokens, model performance shows only negligible changes across different retention ratios. Visual-CLS attention also shows restricted effectiveness as evidenced by the limited improvement of perception scores when the retain ratio increases.

The performance of visual-CLS attention method is limited because the CLS token only aggregates visual information while lacking information from text tokens, which are critical for multimodal understanding. For visual-text attention method, we analyze attention maps from a middle decoder layer, as shown in Figure 2c. Prior works (Wang et al., 2023; Chen et al., 2024a; Huang et al., 2024b) suggests that column-wise attention weights reflect the degree of information aggregated by each token, with tokens that exhibit disproportionately large column-wise attention weights being referred to as "sink tokens". In Figure 2c, it is obvious that text tokens exhibit much lower attention weights than sink tokens. This indicates that text tokens carry limited aggregated information in the middle layers, making visual–text attention ineffective for selecting important visual tokens.

## 4 SINK TOKENS AS AN ANCHOR FOR TOKEN IMPORTANCE ESTIMATION

While most text tokens in intermediate layers receive uniformly low attention, sink tokens attract disproportionately high column-wise attention. Previous work (Huang et al., 2024b) suggests that sink tokens aggregate global information and this aggregation behavior may result in the loss of important visual evidence and potentially lead to hallucinations. To mitigate this issue, they first locate sink tokens and then attempt to relocate attention away from them toward visual tokens. To identify sink tokens, Huang et al. (2024b) computes column-wise attention sums and then selects those exceeding a manually specified threshold. Another location method typically assumes that the first token (for example, BOS) is always the sink token (Xiao et al., 2023). However, such strategies suffer from two limitations: (i) the sink position is not always fixed, as sinks may emerge at different layers or vary across inputs, and (ii) proper thresholds may differ across models or tasks. To overcome these issues, we propose a novel 2-means clustering-based method adaptively partitions tokens into high and low aggregation groups, enabling robust and generalizable sink detection without prior assumptions. (Details are provided in Section 5.1)

To further validate that sink tokens can act as global information aggregators, we compare the effect of removing sink tokens with other tokens on LLaVA-1.5-7B, and find that eliminating sink tokens causes a much larger performance drop. In our settings, the token sequence consists of the first 35 positions as system tokens, followed by visual tokens and text tokens. We first remove the sink

Table 1: Perception score on MME and accuracy on POPE (MS-COCO popular setting) after pruning at the 13th decoder layer in LLaVA-1.5-7B. Sink tokens are first pruned, and subsequently they are retained while other tokens are pruned in a cumulative manner—starting with system tokens, then extending to visual tokens, and finally to text tokens.

| Metric | Baseline | | System Tokens (Total=33) | | | | Sys+Vis Tokens | Text Tokens | |
|---|---|---|---|---|---|---|---|---|---|
| | All | Sink removed | Drop-5 | Drop-15 | Drop-25 | Drop-All | Drop-All | Drop-5 | Drop-10 |
| MME | 1491.56 | 1211.55 | 1489.44 | 1490.33 | 1489.55 | 1485.42 | 1500.69 | 1501.96 | 1467.33 |
| POPE | 86.04 | 50.24 | 84.19 | 84.68 | 85.62 | 85.24 | 85.86 | 84.41 | 70.67 |

tokens, as shown in Table 1, and find that the model performance drops significantly; for example, the accuracy of POPE (Li et al., 2023b) decreases by 35.8%. Conversely, when sink tokens are retained while progressively discarding all system and visual tokens, model performance remains stable or even improves. Although dropping text tokens leads to gradual performance degradation on MME dataset, pruning 10 text tokens still results in 1467.33 points, far higher than 1211.55 points when sink tokens are pruned. These results confirm that LVLMs rely on sink tokens as the primary carriers of global information, whereas most system and visual tokens contribute little. Therefore, by leveraging attention between these sink tokens and visual tokens, we can obtain more reliable importance scores for distinguishing informative visual tokens from redundant ones, as illustrated in Figure 2a and 2b.

## 5 METHOD

Building on the findings above, we propose a novel decoding strategy **ARTS** to mitigate hallucinations in LVLMs. It mainly consists of two steps: (i) localizing sink tokens using 2-means algorithm and (ii) reinjecting original visual embeddings to enhance visual information and utilizing visual-sink cross attention to identify and prune redundant visual tokens in middle decoder layers. The overall framework of ARTS is shown in Figure 3.

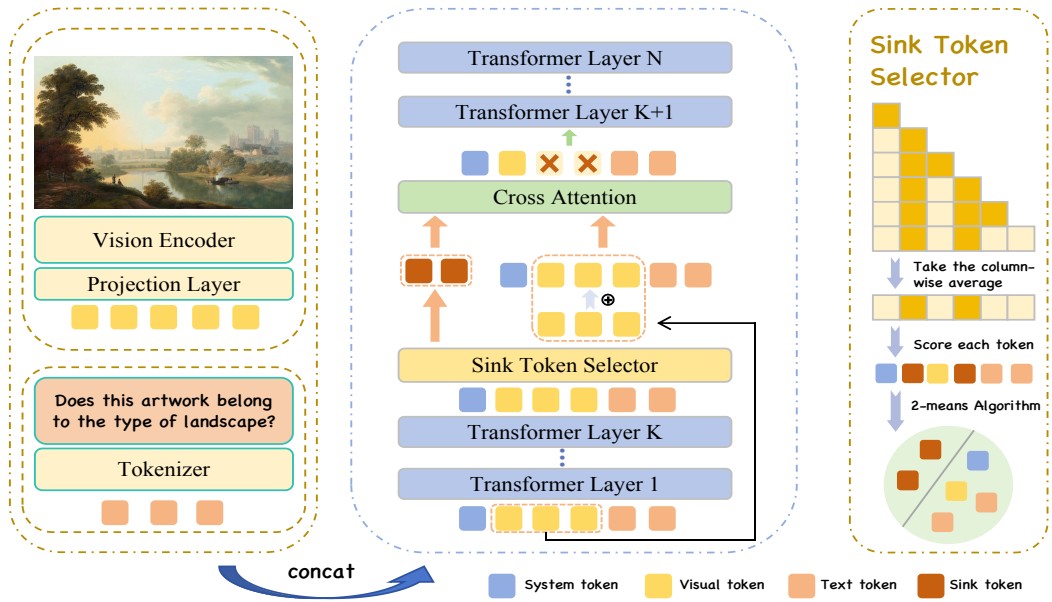

Figure 3: Architecture of the proposed ARTS. At intermediate layer K, sink tokens are first located using the 2-means algorithm, then the original visual information is reinjected. Next, we use cross-attention between updated visual tokens and sink tokens to identify and prune redundant visual tokens.

## 5.1 SINK TOKEN LOCALIZATION

The hidden states at an intermediate decoder layer $K$ are denoted as $H \in \mathbb{R}^{L \times d}$, where $L = L_t + L_v$ is the total number of tokens, consisting of $L_t$ text tokens and $L_v$ visual tokens and $d$ is the hidden dimension. We further define the index sets $\mathcal{T}$ for text positions and $\mathcal{V}$ for visual positions.

We first define $\bar{A}_K \in \mathbb{R}^{L \times L}$ as the head-averaged self-attention map at decoder layer $K$, where each row corresponds to a query position and each column corresponds to a key position. To quantify how strongly each position attracts attention from the entire sequence, we compute the column-wise sum of $\bar{A}_K$, aggregating attention received across all query positions:

$$c_j = \sum_{i=1}^{L} \bar{A}_K[i, j], \qquad j = 1, \ldots, L. \tag{1}$$

We then identify sink positions by running *2-means* on the scalar set $\{c_j\}_{j=1}^{L}$ with Euclidean distance (Details can be found in Appedndix D). Then, we define the set of sink positions $\mathcal{S}$ as those belonging to the cluster with the larger center.

## 5.2 VISUAL INFORMATION ENHANCEMENT AND INFORMATIVE TOKEN SELECTION

Before computing importance scores for visual tokens, we reinforce their representations by reinjecting the original embeddings into the hidden states at layer $K$, thereby restoring visual details that are consistently lost during the aggregation process into sink tokens, as observed in (Huang et al., 2024b). Formally, for each visual token index $i \in \mathcal{V}$, we modify its layer-$K$ hidden state by adding the original visual embedding $V_{\text{orig},i}$:

$$\tilde{H}_i = \begin{cases} H_i + V_{\text{orig},i}, & i \in \mathcal{V}, \\ H_i, & i \in \mathcal{T}, \end{cases}$$

Using these enhanced visual hidden states $\tilde{H}_V$, we then compute the cross-attention between sink tokens $H_S$ and visual tokens $\tilde{H}_V$ at layer $K$ (For notational simplicity and clarity, we omit the linear projections $W_Q$ and $W_K$ and directly use $\tilde{H}_V$ and $H_S$.):

$$\bar{A}_{K,\mathcal{S} \to \mathcal{V}} = \text{softmax}\left(\frac{H_S \tilde{H}_V^\top}{\sqrt{d}}\right) \in \mathbb{R}^{L_s \times L_v},$$

We then compute an importance score for each visual token by averaging its normalized attention weight across all sink tokens. Specifically, for each visual token $i \in \mathcal{V}$ we define:

$$r_i = \frac{1}{|\mathcal{S}|} \sum_{s \in \mathcal{S}} \bar{A}_{K,\mathcal{S} \to \mathcal{V}}[s, i], \qquad \forall\, i \in \mathcal{V},$$

where $r_i$ represents the importance score of token $i$, reflecting the average attention it receives from the sink tokens. Collecting these values in order of visual token indices yields a score vector $r_\mathcal{V} \in \mathbb{R}^{L_v}$.

Finally, given a retain ratio $p \in (0, 1]$, we determine the number of visual tokens to retain as $k = \lfloor p \cdot L_v \rfloor$. We preserve the top-$k$ visual tokens in their original order and discard the rest, while all text tokens $\mathcal{T}$ remain intact.

## 6 EXPERIMENT

### 6.1 SETUP

**Datasets** We evaluate our method on four datasets that explicitly target hallucination in LVLMs. MME (Fu et al., 2024) provides a comprehensive evaluation suite of 14 tasks, which are grouped into perception and cognition to measure both comprehensive hallucination and higher-level reasoning. Each task contributes up to 200 points, resulting in maximum scores of 2000 for perception and 800

for cognition. POPE (Li et al., 2023b) introduces a scalable framework for detecting object hallucinations, leveraging SEEM-based annotations on MS-COCO (Lin et al., 2014), A-OKVQA Schwenk et al. (2022) and GQA (Hudson & Manning, 2019). In addition, we evaluate the generalization of our method on two real-world VQA benchmarks: VizWiz (Gurari et al., 2018), which contains noisy and ambiguous images, and MM-Vet (Yu et al., 2023), which assesses multimodal models across six core capabilities—recognition, knowledge, spatial awareness, language generation, OCR, and math—with GPT-4 (Achiam et al., 2023) used as the automatic evaluator. Together, these datasets provide a comprehensive testbed for evaluating hallucinations of LVLMs in open-domain settings.

**Models and Baselines** To ensure generality, we evaluate our method across four representative 7B-level LVLMs: LLaVA-1.5 (Liu et al., 2023b), mPLUG-Owl2 (Ye et al., 2024b), InstructBLIP Dai et al. (2023), and MiniGPT-4 (Zhu et al., 2023). In addition, we also conduct experiments on LLaVA-1.5-13B to examine scalability across different model sizes, which is reported in Appendix A.3. For baseline comparisons, we evaluate ARTS against six representative decoding methods, including VCD (Leng et al., 2024), DoLa (Chuang et al., 2023), DAMO (Wang et al., 2025), SID (Huo et al., 2024), and OPERA (Huang et al., 2024b), SPIN (Sarkar et al., 2025). All baselines are implemented with their official configurations. For OPERA, we reduce the beam size to 4 due to computational constraints while keeping all other settings identical to its official configuration. Moreover, all methods are evaluated under identical conditions with the decoding temperature fixed at zero for fair comparison. The detailed parameter settings of ARTS are provided in appendix C.

## 6.2 RESULTS

**Results on Comprehensive Hallucination Dataset** We assess the capability of ARTS to mitigate hallucinations using the perception subset of the MME benchmark, which consists of ten tasks specifically designed to evaluate hallucination. The cognition subset targets higher-level reasoning, and its results are deferred to Appendix A.2. To ensure a thorough and fair evaluation, we conduct experiments across multiple representative LVLMs. As shown in Table 6, ARTS consistently surpasses all baseline decoding strategies, achieving the highest overall scores on every model. Notably, it improves LLaVA-1.5-7B from 1491.56 to 1520.68 and mPLUG-Owl2 from 1459.54 to 1474.29, and achieves the largest relative gain on InstructBLIP with an increase of 63.22 points. On MiniGPT-4, ARTS significantly surpasses the strongest decoding method SID by 14.66 points. These results demonstrate that ARTS achieves consistent gains across diverse LVLMs, confirming its effectiveness in reducing hallucinations.

Table 2: Experimental results on MME dataset across LLaVA1.5, mPLUG-Owl2, InstructBLIP, MiniGPT4. (The total score is 2000)

| Model | Vanilla | VCD | DAMO | DOLA | SID | OPERA | SPIN | ARTS |
|---|---|---|---|---|---|---|---|---|
| LLaVA-1.5 | 1491.56 | 1484.96 | 1513.51 | 1495.02 | 1513.85 | 1510.68 | 1500.13 | **1520.68** |
| mPLUG-Owl2 | 1459.54 | 1311.52 | 1462.95 | 1456.31 | 1467.43 | 1464.81 | 1456.96 | **1474.29** |
| InstructBLIP | 1271.54 | 1162.34 | 1298.76 | 1274.25 | 1295.76 | 1290.82 | 1287.18 | **1334.76** |
| MiniGPT4 | 731.87 | 725.65 | 737.26 | 742.23 | 749.72 | 744.23 | 739.08 | **764.38** |

**Results on Object Hallucination Dataset** To assess the effectiveness of our method in mitigating object-level hallucinations, we conduct experiments on the SEEM-annotated versions of MS-COCO, A-OKVQA, and GQA, as provided by the POPE benchmark. Following standard practice, we report two metrics, accuracy and F1 score, which jointly capture the consistency of object recognition across different models and decoding approaches. The results are summarized in Table 3, it is clear that ARTS consistently outperforms both vanilla decoding and strong baselines across datasets. On MiniGPT4 under the random setting, it improves MSCOCO F1 from 59.94% to 63.99%, a gain of 4.05%, clearly surpassing DAMO by 3.34%. On LLaVA-1.5-7B under the random setting, ARTS raises the F1 score in GQA from 87. 22% with Vanilla decoding to 88. 24%, surpassing the strongest baseline SID by 0.83%. These results underscore the superior performance of ARTS in mitigating object hallucinations.

**Results on Generalized Dataset** As shown in Table **??**, ARTS achieves competitive results on two generalized datasets. On VizWiz, it reaches 50.42% overall accuracy and highest 75.16% in

Table 3: Experimental results of various decoding strategies on the SEEM-annotated MSCOCO,A-OKVQA and GQA datasets from POPE using four models: LLaVA-1.5 and MiniGPT4. The best values for each metric across all models and decoding strategies are highlighted in **bold.**

| Setting | Model | Decoding | MSCOCO | | A-OKVQA | | GQA | |
|---|---|---|---|---|---|---|---|---|
| | | | Accuracy | F1 Score | Accuracy | F1 Score | Accuracy | F1 Score |
| Random | LLaVA-1.5 | Vanilla | 89.21 | 89.41 | 87.12 | 88.21 | 86.01 | 87.22 |
| | | VCD | 88.47 | 87.56 | 86.01 | 85.35 | 84.10 | 85.23 |
| | | DoLa | 89.13 | 88.23 | 86.24 | 87.57 | 85.29 | 86.81 |
| | | DAMO | **90.10** | 89.49 | 87.31 | 88.59 | 86.47 | 87.03 |
| | | OPERA | 89.61 | 89.03 | 87.04 | 88.33 | 86.22 | 87.21 |
| | | SID | 90.02 | 89.32 | 87.19 | 88.68 | 86.32 | 87.41 |
| | | **ARTS** | 89.94 | **89.90** | **87.50** | **89.02** | **87.03** | **88.24** |
| | MiniGPT4 | Vanilla | 70.35 | 59.94 | 69.79 | 61.41 | 70.14 | 63.11 |
| | | VCD | 68.43 | 57.76 | 67.62 | 59.28 | 67.13 | 60.69 |
| | | DoLa | 70.44 | 59.91 | 67.21 | 60.21 | 68.34 | 62.21 |
| | | DAMO | 71.01 | 60.65 | 70.82 | 62.97 | 70.35 | 63.21 |
| | | OPERA | 69.15 | 59.11 | 70.41 | 62.82 | 69.38 | 62.16 |
| | | SID | 71.53 | 61.27 | 71.03 | 62.59 | 70.44 | 63.03 |
| | | **ARTS** | **72.21** | **63.99** | **71.76** | **65.36** | **70.52** | **63.62** |
| Popular | LLaVA-1.5 | Vanilla | 86.04 | 86.13 | 80.10 | 83.02 | 74.21 | 79.01 |
| | | VCD | 84.45 | 85.81 | 78.74 | 81.78 | 70.76 | 76.28 |
| | | DoLa | 84.26 | 85.97 | 79.27 | 82.62 | 72.23 | 78.50 |
| | | DAMO | 86.25 | 86.67 | 79.78 | 83.05 | 75.14 | 80.02 |
| | | OPERA | 86.28 | 86.55 | 80.02 | 83.15 | 74.13 | 79.24 |
| | | SID | 86.21 | 86.24 | 80.74 | 83.21 | 74.76 | 79.52 |
| | | **ARTS** | **86.93** | **87.35** | **80.80** | **83.50** | **76.04** | **80.88** |
| | MiniGPT4 | Vanilla | 68.23 | 59.02 | 65.46 | 58.55 | 63.39 | 57.79 |
| | | VCD | 65.62 | 56.12 | 62.38 | 55.34 | 62.17 | 56.58 |
| | | DoLa | 67.15 | 57.37 | 63.33 | 56.62 | 61.02 | 55.45 |
| | | DAMO | 68.31 | 60.03 | 66.04 | 59.98 | 63.21 | 57.81 |
| | | OPERA | 67.33 | 58.74 | 64.49 | 58.01 | 63.12 | 57.44 |
| | | SID | **68.43** | 60.08 | 66.63 | 60.31 | 64.02 | 58.22 |
| | | **ARTS** | 68.32 | **60.87** | **67.26** | **61.75** | **64.24** | **58.78** |
| Adversarial | LLaVA-1.5 | Vanilla | 79.07 | 81.15 | 69.03 | 75.30 | 66.19 | 72.12 |
| | | VCD | 74.37 | 77.32 | 66.24 | 70.13 | 65.23 | 70.45 |
| | | DoLa | 77.21 | 79.32 | 68.82 | 75.21 | 66.28 | 71.42 |
| | | DAMO | 79.79 | 81.32 | 69.13 | 76.02 | 67.61 | 73.34 |
| | | OPERA | 79.23 | 81.18 | 69.14 | 75.58 | 67.13 | 73.33 |
| | | SID | 79.67 | 81.24 | 69.32 | 75.46 | 67.51 | 73.83 |
| | | **ARTS** | **80.87** | **82.50** | **69.83** | **76.30** | **68.76** | **75.69** |
| | MiniGPT4 | Vanilla | 67.53 | 58.30 | 62.06 | 55.91 | 62.68 | 57.60 |
| | | VCD | 65.13 | 55.62 | 60.86 | 52.58 | 59.48 | 54.27 |
| | | DoLa | 66.48 | 56.39 | 61.73 | 55.67 | 61.79 | 56.62 |
| | | DAMO | 67.28 | 58.14 | 61.48 | 54.87 | 62.99 | 58.01 |
| | | OPERA | 67.57 | 58.49 | 62.01 | 56.24 | 62.84 | 57.40 |
| | | SID | 67.85 | 59.17 | 62.05 | 56.68 | 63.04 | 57.92 |
| | | **ARTS** | **68.35** | **60.30** | **62.86** | **58.23** | **63.18** | **58.13** |

the Unanswerable category. On MM-Vet, it obtains a total score of 29.2%, with 21.0% in OCR and 24.4% in Spatial Reasoning. Compared with alternative decoding strategies, ARTS maintains robust and balanced performance across various tasks, underscoring its strong generalizability.

# 7 ABLATION STUDIES

## 7.1 EFFECT OF REINJECTING ORIGINAL VISUAL EMBEDDINGS

We compare pruning with and without reinjecting the original visual embeddings on the MME perception task using LLaVA-1.5-7B. As shown in Figure 4a, reinjection consistently achieves higher scores across retention ratios, with the largest margin at $P = 0.3$, where it reaches 1520.7 compared to 1510.6 without reinjection. These results indicate that restoring original visual embeddings before pruning helps the model better preserve essential information.

Table 4: Evaluation of ARTS and other decoding methods on LLaVA-1.5 using VizWiz and MM-Vet.

| Decoding | VizWiz | | | | | MM-Vet | | | | | | |
|---|---|---|---|---|---|---|---|---|---|---|---|---|
| | Number | Yes/No | Unans. | Other | **Overall** | Rec | OCR | Know | Gen | Spat | Math | **Total** |
| Vanilla | 47.62 | 78.26 | 74.30 | 38.11 | 50.05 | 33.2 | 20.0 | 15.6 | 18.0 | 22.3 | **11.5** | 28.4 |
| VCD | 42.70 | 77.64 | 72.54 | 38.12 | 49.47 | 26.2 | 19.9 | 9.6 | 8.1 | 22.0 | 7.7 | 24.6 |
| DoLa | **53.33** | **80.00** | 69.64 | 37.36 | 48.43 | 32.3 | 19.4 | **16.8** | 14.9 | 23.5 | 7.7 | 28.5 |
| DAMO | 48.10 | 78.88 | 71.78 | 36.63 | 48.41 | 32.3 | 18.2 | 16.1 | 14.9 | 22.4 | 7.7 | 27.8 |
| **ARTS** | 49.05 | 77.87 | **75.16** | **38.30** | **50.42** | 32.7 | **21.0** | 16.7 | 15.5 | **24.4** | **11.5** | **29.2** |

## 7.2 SENSITIVITY OF HYPERPARAMETERS

We evaluate the sensitivity of the retain ratio $P$ and the starting layer $K$ on the MME perception benchmark (Fu et al., 2024) using LLaVA-1.5. As shown in Figure 4b, pruning visual tokens from earlier layers (e.g., $K = 11$) leads to a clear performance drop, with scores decreasing as the retention ratio decreases and reaching a minimum of 1255.9 at a ratio of 0.2. This is because visual tokens at shallow layers still contain rich visual information that is critical for subsequent reasoning. In contrast, pruning in the middle layers ($K = 12$–$14$) consistently improves performance across different retain ratios, suggesting that redundant information accumulates at this stage. For deeper layers ($K = 15$–$16$), the performance remains slightly lower than baseline across different ratios, indicating that redundant information has already fully flowed into text tokens. These findings highlight that moderate pruning in the mid-layers is most effective, while early pruning is detrimental and late pruning has little impact on overall model performance.

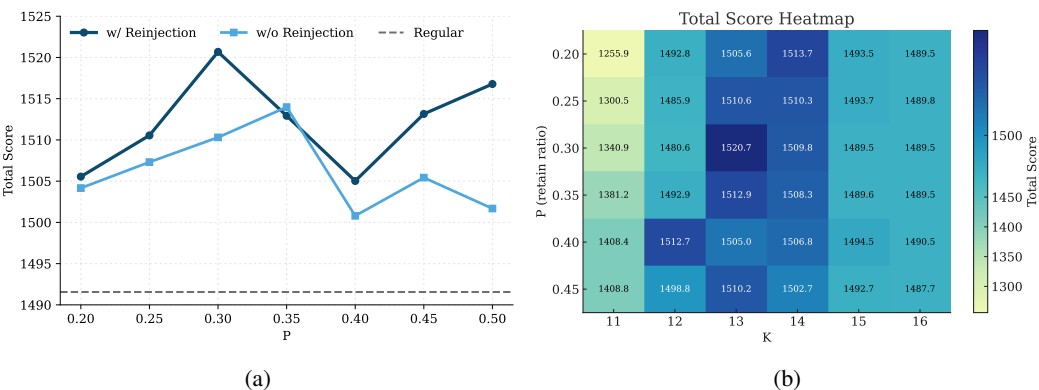

(a)                                                    (b)

Figure 4: (a) Sensitivity of pruning performance to retain ratio $P$ and starting layer $K$ on the MME perception task using LLaVA-1.5-7B. (b) Effect of visual token reinjection on the MME perception task using LLaVA-1.5-7B.

## 8 CONCLUSION

In this paper, we introduced **ARTS**, a training-free approach to alleviating hallucinations in Large Vision-Language Models (LVLMs) by selectively retaining informative visual tokens at intermediate decoder layers. Our analysis revealed that, contrary to prior assumptions, most visual tokens at this stage become redundant or even misleading, and their removal can improve semantic coherence and reduce hallucinations. By reinjecting original visual information and leveraging sink–visual attention, ARTS effectively preserves essential visual information while reducing inherent redundancy. Extensive experiments across different LVLM architectures demonstrate that ARTS consistently improves factual accuracy and robustness, underscoring its potential to enhance the reliability and trustworthiness of multimodal systems.

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

# A  MORE EXPERIMENTS

## A.1  EXTRA POPE RESULTS ON MPLUG-OWL2 AND INSTRUCTBLIP

We further evaluate our ARTS on mPLUG-Owl2 and InstructBLIP using POPE dataset. As shown in Table 5, our method nearly outperforms all baselines across MSCOCO, A-OKVQA, and GQA under the three settings, demonstrating not only robustness to different evaluation protocols but also generalizability across distinct LVLM architectures.

Table 5: Experimental results of various decoding strategies on the SEEM-annotated MSCOCO,A-OKVQA and GQA datasets from POPE using four models: InstructBLIP and mPLUG-Owl2. The best values for each metric across all models and decoding strategies are highlighted in **bold**.

| Setting | Model | Decoding | MSCOCO | | A-OKVQA | | GQA | |
|---|---|---|---|---|---|---|---|---|
| | | | Accuracy | F1 Score | Accuracy | F1 Score | Accuracy | F1 Score |
| Random | mPLUG-Owl2 | Vanilla | 86.44 | 85.13 | 87.12 | 86.77 | 86.10 | 85.13 |
| | | VCD | 85.17 | 84.42 | 85.56 | 85.23 | 84.43 | 82.87 |
| | | DoLa | 86.62 | 85.23 | 86.58 | 85.68 | 86.21 | 84.19 |
| | | DAMO | 87.74 | 86.68 | 88.14 | 87.82 | 86.92 | 85.85 |
| | | OPERA | 87.32 | 86.14 | 87.91 | 87.91 | 86.31 | 85.17 |
| | | SID | 87.19 | 86.33 | 87.98 | 88.14 | **87.01** | 86.04 |
| | | **ARTS** | **87.97** | **87.06** | **88.90** | **88.78** | 86.90 | **86.18** |
| | InstructBLIP | Vanilla | 87.23 | 85.84 | 88.51 | 88.18 | 86.12 | 87.16 |
| | | VCD | 84.24 | 83.09 | 83.44 | 82.13 | 81.24 | 82.23 |
| | | DoLa | 87.17 | 85.89 | 86.64 | 87.01 | 84.21 | 85.17 |
| | | DAMO | 87.99 | 86.98 | 87.21 | 87.45 | 86.76 | 87.19 |
| | | OPERA | 87.74 | 86.23 | 87.02 | 87.13 | 86.13 | 86.79 |
| | | SID | 88.03 | 86.90 | 87.85 | 88.01 | 86.65 | 87.14 |
| | | **ARTS** | **88.60** | **87.76** | **88.93** | **89.06** | **87.33** | **87.47** |
| Popular | mPLUG-Owl2 | Vanilla | 84.43 | 84.12 | 83.72 | 84.02 | 79.04 | 80.13 |
| | | VCD | 82.28 | 82.14 | 80.23 | 80.43 | 77.13 | 78.34 |
| | | DoLa | 83.87 | 82.16 | 82.24 | 82.89 | 80.01 | **80.78** |
| | | DAMO | 85.02 | 85.23 | 83.87 | 84.19 | 79.03 | 79.82 |
| | | OPERA | 84.98 | 84.89 | 83.89 | 83.87 | **80.11** | 80.27 |
| | | SID | 85.17 | 85.02 | 83.77 | 83.35 | 79.34 | 79.14 |
| | | **ARTS** | **86.97** | **86.21** | **84.47** | **84.97** | 79.37 | 79.83 |
| | InstructBLIP | Vanilla | 83.40 | 84.12 | 78.34 | 81.92 | 76.25 | 79.01 |
| | | VCD | 80.23 | 80.04 | 76.14 | 78.34 | 73.27 | 75.88 |
| | | DoLa | 82.39 | 83.41 | 78.85 | 80.42 | 74.82 | 77.21 |
| | | DAMO | 84.21 | 84.27 | 79.01 | 81.34 | 75.31 | 78.41 |
| | | OPERA | 83.71 | 84.08 | 78.86 | 81.27 | 75.62 | 78.57 |
| | | SID | 84.16 | 84.31 | 79.09 | 81.63 | 75.79 | 78.37 |
| | | **ARTS** | **85.23** | **84.70** | **80.73** | **82.37** | **77.77** | **79.90** |
| Adversarial | mPLUG-Owl2 | Vanilla | 80.91 | 81.31 | 76.28 | 77.21 | 77.24 | 79.02 |
| | | VCD | 78.31 | 80.10 | 75.34 | 74.42 | 75.31 | 77.46 |
| | | DoLa | 79.24 | 80.13 | 75.28 | 75.35 | 75.16 | 78.82 |
| | | DAMO | 82.14 | 82.33 | 76.73 | 78.01 | 78.42 | 80.13 |
| | | OPERA | 81.42 | 81.90 | 76.75 | 77.82 | 78.79 | 80.22 |
| | | SID | 82.08 | 82.21 | 76.47 | 78.31 | 78.69 | 80.19 |
| | | **ARTS** | **83.87** | **83.37** | **77.57** | **79.65** | **79.67** | **80.60** |
| | InstructBLIP | Vanilla | 80.91 | 81.98 | 71.56 | 77.28 | 71.42 | 75.94 |
| | | VCD | 78.24 | 78.67 | 71.02 | 75.78 | 70.25 | 74.26 |
| | | DoLa | 79.21 | 80.02 | 70.22 | 76.56 | 69.14 | 74.44 |
| | | DAMO | 81.85 | 81.77 | 71.42 | 77.01 | 71.56 | 76.11 |
| | | OPERA | 81.21 | 81.75 | 70.83 | 76.16 | 71.12 | 75.75 |
| | | SID | 81.17 | 81.68 | 71.78 | 76.78 | 72.22 | 76.12 |
| | | **ARTS** | **82.37** | **82.27** | **73.60** | **77.33** | **73.76** | **77.12** |

## A.2  EXPERIMENT RESULTS ON HIGH-LEVEL REASONING TASK

We evaluate the cognition subset to assess high-level reasoning capability of ARTS. On LLaVA-1.5-7B, ARTS achieves the highest score, exceeding the strongest baseline DOLA by 14.21 points. On mPLUG-Owl2, while all baseline methods exhibit notable performance drops, ARTS maintains

the original model performance, demonstrating its robustness in preserving reasoning ability under challenging conditions.

Table 6: Experimental results on MME dataset across LLaVA1.5-7b, mPLUG-Owl2. (The total score is 800)

| Model | Vanilla | VCD | DAMO | DOLA | SID | OPERA | SPIN | ARTS |
|---|---|---|---|---|---|---|---|---|
| LLaVA1.5-7b | 294.29 | 287.50 | 305.25 | 318.21 | 311.42 | 299.98 | 295.71 | **332.42** |
| mPLUG-Owl2 | **345.71** | 329.29 | 332.86 | 265.00 | 337.14 | 332.28 | 332.56 | 345.24 |

## A.3 EXPERIMENT RESULTS ON LLAVA1.5-13B

We evaluate our **ARTS** on the MME dataset using LLaVA1.5-13B, as shown in Table 7. We perform pruning at layer 17 with a retain ratio of 0.5. ARTS achieves the highest overall score (1517.22), consistently surpassing strong baselines such as VCD (1504.44) and DCLA (1504.82). Compared with other decoding strategies that only achieve gains in a few categories (e.g., VCD in *Count* or DAMO in *Scene*), ARTS delivers improvements that are more evenly distributed across all evaluation dimensions, indicating that our method effectively reduces hallucinations and enhances factual alignment.

Table 7: Experimental results of various decoding strategies on MME dataset using LLaVA1.5-13b.

| Model | Decoding | Existence | Count | Position | Color | Posters | Celebrity | Scene | Landmark | Artwork | OCR | Total |
|---|---|---|---|---|---|---|---|---|---|---|---|---|
| | Regular | 188.33 | 145.00 | 123.33 | 160.00 | 159.52 | 159.71 | 157.25 | 141.75 | 121.75 | 147.50 | 1504.15 |
| | VCD | 190.00 | 163.33 | 120.00 | 175.00 | 151.70 | 159.41 | 158.25 | 129.00 | 125.25 | 132.50 | 1504.44 |
| LLaVA1.5-13b | DoLa | 190.00 | 150.00 | 123.33 | 160.00 | 160.54 | 157.06 | 155.75 | 134.50 | 124.00 | 147.50 | 1502.69 |
| | DAMO | 190.00 | 125.00 | 113.33 | 150.00 | 166.66 | 152.06 | 163.25 | 166.50 | 107.75 | 140.00 | 1474.56 |
| | DCLA | 188.33 | 145.00 | 123.33 | 160.00 | 159.52 | 160.88 | 158.00 | 140.50 | 121.75 | 147.50 | 1504.82 |
| | **ARTS** | 190.00 | 155.00 | 121.67 | 170.00 | 159.52 | 158.53 | 156.50 | 134.50 | 124.00 | 147.50 | **1517.22** |

# B DETAILS OF TWO TOKEN-IMPORTANCE METRICS

## B.1 DETAILS OF VISUAL-TEXT ATTENTION

Let $H_v \in \mathbb{R}^{L_v \times d}$ denote the visual-token embeddings, and $H_q \in \mathbb{R}^{L_q \times d}$ and $H_{\text{resp}} \in \mathbb{R}^{L_o \times d}$ the embeddings of the input query and generated response, respectively. We form the textual sequence by concatenation

$$\tilde{H}_t := \begin{bmatrix} H_q \\ H_{\text{resp}} \end{bmatrix} \in \mathbb{R}^{(L_q + L_o) \times d}.$$

For notational simplicity and clarity, we omit the linear projections $W_Q$ and $W_K$ and directly use $\tilde{H}_t$ and $H_v$. At decoder layer $K$, consider the cross-attention from textual queries to visual keys produced is

$$A^{(K)} = \text{softmax}\left(\frac{\tilde{H}_t H_v^\top}{\sqrt{d}}\right) \in \mathbb{R}^{(L_q + L_o) \times L_v},$$

The importance of a visual token $i$ is defined as its average attention mass over all textual positions:

$$r_i = \frac{1}{L_q + L_o} \sum_{j=1}^{L_q + L_o} A_{j,i}^{(K-1)}, \qquad i = 1, \dots, L_v.$$

## B.2 DETAILS OF VISUAL-CLS ATTENTION

Let $H_v \in \mathbb{R}^{L_v \times d}$ denote the visual-token embeddings produced by the vision encoder, which contains a designated [CLS] token at index $i_{\text{cls}}$. We extract this representation as

$$h_{\text{cls}} = H_v[i_{\text{cls}}, :] \in \mathbb{R}^d.$$

For notational simplicity and clarity, we omit the linear projections $W_Q$ and $W_K$ and directly use $h_{\text{cls}}$ and $H_v$. At decoder layer $K$, consider the cross-attention from the `[CLS]` query to all visual tokens, computed as

$$a^{(K)} \;=\; \text{softmax}\left(\frac{h_{\text{cls}} H_v^\top}{\sqrt{d}}\right) \;\in\; \mathbb{R}^{L_v},$$

The importance of a visual token $i$ is then defined as its attention mass with respect to the `[CLS]` query:

$$r_i \;=\; a_i^{(K)}, \qquad i = 1, \dots, L_v.$$

## C  PARAMETER SETTINGS

We provide our detailed hyperparameter settings in Table 8. For each LVLM, we report the starting layer $K$ and the retention ratio $P$ used throughout all experiments. Here, "m" denotes the configuration adopted for evaluations on the MME benchmark, while "p" corresponds to the configuration used for the POPE benchmark. Moreover, even with non-optimal $(K, P)$ values, our method consistently outperforms the regular decoding baseline, highlighting its robustness and general applicability.

Table 8: Hyperparameter settings for different LVLMs. $K$ denotes the starting layer and $P$ the retention ratio.

|   | LLaVA1.5-7B m/p | mPLUG-Owl2 m/p | InstructBLIP m | InstructBLIP p | MiniGPT4 m | MiniGPT4 p |
|---|---|---|---|---|---|---|
| $K$ | 13 | 14 | 15 | 16 | 16 | 17 |
| $P$ | 0.30 | 0.35 | 0.20 | 0.50 | 0.05 | 0.50 |

## D  2-MEANS CLUSTERING WITH EUCLIDEAN DISTANCE

**A. Global Minimization Objective**

Given a set of scalar values $\{x_i\}_{i=1}^N$, 2-means clustering with Euclidean distance aims to partition them into two clusters $S_1$ and $S_2$ by minimizing the total within-cluster variance:

$$\min_{\mu_1, \mu_2, S_1, S_2} \sum_{i \in S_1} (x_i - \mu_1)^2 + \sum_{i \in S_2} (x_i - \mu_2)^2. \tag{2}$$

Here,

- $x_i$ denotes the $i$-th scalar value (in our method, an attention-derived scalar score),

- $S_1, S_2$ are the two clusters (index sets of samples),

- $\mu_1, \mu_2$ are the corresponding cluster means.

This objective encourages values within each cluster to stay close to their cluster mean (reducing intra-cluster variance) and naturally separates the set into a "large-value" cluster and a "small-value" cluster.

**B. Iterative Alternating Minimization Procedure**

The above objective is typically solved by the standard alternating minimization procedure of k-means:

**1. Initialization**  The two cluster means $\mu_1$ and $\mu_2$ are first initialized, e.g., by randomly selecting two values from $\{x_i\}$ or using other standard initialization strategies.

**2. Assignment Step**  Given the current means $\mu_1$ and $\mu_2$, each value $x_i$ is assigned to the cluster with the closest mean:

$$S_1 = \{\, i : |x_i - \mu_1| \le |x_i - \mu_2|\,\}, \qquad S_2 = \{\, i : |x_i - \mu_1| > |x_i - \mu_2|\,\}. \tag{3}$$

Using $|x_i - \mu_1|$ and $|x_i - \mu_2|$ (instead of squared distance) is valid, because in one dimension $(x_i - \mu)^2$ and $|x_i - \mu|$ are monotonically equivalent, and thus yield the same $\arg\min$.

**3. Update Step**  After the assignment step, the new cluster means are computed as the average of all values assigned to each cluster:

$$\mu_1^{\text{new}} = \frac{1}{|S_1|} \sum_{i \in S_1} x_i, \qquad \mu_2^{\text{new}} = \frac{1}{|S_2|} \sum_{i \in S_2} x_i. \tag{4}$$

The assignment and update steps are repeated until the cluster memberships no longer change, yielding a locally optimal partition of the values into two groups.

# E  COMPUTATIONAL EFFICIENCY

In this section, we first report the memory usage of various hallucination mitigation methods to highlight the lightweight nature of our approach. We then compare ARTS with two other token pruning strategies in terms of GPU memory consumption, FLOPs, and inference latency, to demonstrate the quality–efficiency trade-off achieved by ARTS.

Despite introducing some additional computation during sink-token detection and redundancy scoring, ARTS significantly reduces the overall computational load through token pruning, resulting in a net efficiency gain. As shown in Table 9, ARTS is the only method that consumes less GPU memory than the base model across both LLaVA-1.5-7B and mPLUG-Owl2.

Table 9: Memory consumption (in GiB) of different hallucination mitigation methods during inference on MME benchmark.

| Model | Vanilla | VCD | DOLA | OPERA | DAMO | ARTS |
|---|---|---|---|---|---|---|
| LLAVA-1.5-7B (GiB) | 15.5 | 16.6 | 15.9 | 23.2 | 15.5 | **15.3** |
| mPLUG-Owl2 (GiB) | 17.2 | 17.6 | 17.3 | 25.1 | 17.2 | **17.1** |

Compared with Visual-CLS and Visual-Text, ARTS incurs only marginal increases in FLOPs, latency, and other computational costs, as shown in Table 10. This demonstrates that ARTS offers a favorable efficiency–performance trade-off.

Table 10: Inference efficiency comparison between Visual-CLS attention, Visual-Text attnention and ARTS with LLaVA-1.5-7B on MME benchmark.

| Retain Ratio | Method | Avg. FLOPs (T) | GPU Mem (GiB) | Latency (ms) | Throughput (it/s) |
|---|---|---|---|---|---|
| 1 | LLaVA-1.5-7B | 8.82 | 15.47 | 138.69 | 7.21 |
| 0.5 | Visual-CLS | 6.40 | 15.18 | 133.73 | 7.48 |
| | Visual-Text | 6.38 | 14.79 | 134.88 | 7.41 |
| | **ARTS** | 6.57 | 15.34 | 136.21 | 7.34 |
| 0.1 | Visual-CLS | 4.76 | 15.05 | 123.06 | 8.12 |
| | Visual-Text | 4.74 | 14.67 | 125.59 | 7.96 |
| | **ARTS** | 4.92 | 15.19 | 127.56 | 7.84 |
| 0.05 | Visual-CLS | 4.59 | 14.74 | 122.38 | 8.17 |
| | Visual-Text | 4.56 | 14.62 | 124.94 | 8.00 |
| | **ARTS** | 4.79 | 15.12 | 127.57 | 7.84 |

# F    CASE STUDY

Here, we provide a qualitative example from the MM-Vet dataset to illustrate the output of ARTS. As shown in Figure F, ARTS corrects a hallucinated response produced by LLaVA-1.5-7B and accurately identifies that Number 2 in the image indicates the mantle.

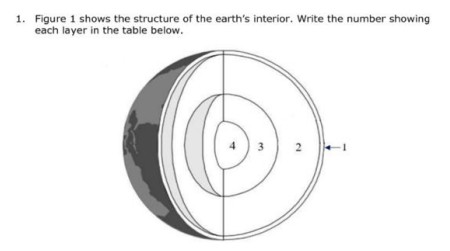

**"What earth's interior structure does number 2 indicate?"**

**LLaVA1.5:**
"Number 2 in the image indicates the outer core of the Earth's interior. The outer core is the layer of the Earth that lies between the mantle and the core, and it is primarily composed of liquid iron and nickel. It is responsible for generating the Earth's magnetic field and plays a crucial role in the Earth's overall structure and functioning."

**ARTS:**
"Number 2 in the image indicates the mantle, which is the layer of the Earth's interior that lies between the core and the crust. The mantle is primarily composed of rock and is responsible for the Earth's tectonic activity, including plate tectonics and the formation of mountains. It is also the layer that transfers heat from the Earth's interior to the surface, contributing to the Earth's climate."

Figure 5: Case Study from MMVet on LLaVA-1.5-7B. Hallucinations are marked in red

