# OpenReview forum: "ARTS: Alleviating Hallucinations in Large Vision–Language Models via Redundancy-Aware Token Selection"
_ICLR.cc/2026/Conference — Submitted to ICLR 2026_

### Official Review · Reviewer_YxSQ · 2025-10-20

**Soundness:** 3
**Presentation:** 3
**Contribution:** 2
**Rating:** 4
**Confidence:** 5

**Summary:**

This paper addresses the hallucination problem in Large Vision-Language Models (LVLMs) from the perspective of visual token redundancy. It proposes ARTS, a decoding-time method that reintegrates original visual embeddings to enrich essential visual information and then leverages sink-token-based attention to identify and retain informative visual tokens in intermediate decoder layers. Extensive experiments demonstrate the effectiveness of ARTS in reducing hallucinations and improving factual alignment across multiple LVLMs and benchmarks.

**Strengths:**

- The paper is clearly written, logically organized, and supported by well-designed figures that effectively illustrate both the motivation and the method.
- ARTS leverages sink tokens as reliable anchors to estimate token importance, offering a principled and adaptive way to identify redundant visual tokens without additional training.
- Extensive experiments on multiple LVLMs and benchmarks show consistent gains over various decoding-based baselines.

**Weaknesses:**

- **Incomplete baseline comparison**: Several recent hallucination mitigation approaches, such as VTI [1], VASparse [2], and CMI-VLD [3], are not included in the comparison. These methods respectively address latent-space steering, vision-aware decoding, and adaptive cross-modal consistency. Including them would strengthen the empirical validation and contextual positioning of this work. In addition, OPERA and SID are missing in Table 7 without explanation.
- **Experimental Models**: The evaluation focuses on relatively early LVLMs (e.g., LLaVA-1.5, InstructBLIP), whereas more recent and capable models such as Qwen2.5-VL or LLaVA-NEXT are not considered. This limits the assessment of scalability and relevance to current-generation systems.
- **Restricted benchmark scope**: The evaluation could be more comprehensive by including additional hallucination-oriented benchmarks such as CHAIR [4], MMBench [5], or the GPT-4-assisted hallucination benchmark [6], which would provide stronger evidence of robustness and generalization.
- **Hyperparameter sensitivity**: ARTS depends on several hyperparameters (e.g., starting layer K, retention ratio P), which appear to require model- and dataset-specific tuning. This sensitivity may limit generalizability and increase the deployment cost across different LVLMs.
- **Time and computational consumption**: As a decoding-time method, a key concern lies in the trade-off between performance gain and computational overhead. The steps of sink-token localization and redundant-token pruning may introduce additional latency and memory cost. Reporting metrics such as inference time, FLOPs, or speedup ratios would help quantify the quality–efficiency trade-off of ARTS.
- **Clarity in Comparison**: Some baseline implementations (e.g., OPERA with reduced beam size) may not fully reflect their best-reported performance, potentially affecting fairness.
- **Missing ablation**: The effect of the number of clusters in the sink-token localization step is not discussed. Such an ablation would clarify whether two clusters are optimal or if finer-grained partitioning could yield additional benefits.

[1]Liu Sheng, et al. Reducing hallucinations in vision-language models via latent space steering. In ICLR 2025.

[2]Zhuang Xianwei, et al. Vasparse: Towards efficient visual hallucination mitigation for large vision-language model via visual-aware sparsification. In CVPR 2025.

[3]Fang Hao, et al. Grounding Language with Vision: A Conditional Mutual Information Calibrated Decoding Strategy for Reducing Hallucinations in LVLMs. In NIPS 2025.

[4]Rohrbach, Anna, et al. Object hallucination in image captioning. In EMNLP 2018.

[5]Liu Yuan, et al. Mmbench: Is your multi-modal model an all-around player? In ECCV 2024.

[6]Zhao, Zhiyuan, et al. Beyond hallucinations: Enhancing lvlms through hallucination-aware direct preference optimization. arXiv preprint arXiv:2311.16839 (2023).

**Questions:**

Refer to Weaknesses.

---

> ### Author Response · Authors · 2025-11-24
>
> We sincerely appreciate your review and the valuable suggestions provided.
> ---
>
> ## **W1: Recent hallucination mitigation approaches are not included in the comparison & missing OPERA and SID on LLaVA-1.5-13B**
>
> 1. Here we compare the performance of **VTI [1]**, **VASparse [2]**, and **CMI-VLD [3]** with our proposed **ARTS** on the **MME Perception benchmark** using **LLaVA-1.5-7B**. As shown in the table below, **ARTS consistently outperforms all three methods**.
>
> | Method       | LLaVA-1.5-7B | MiniGPT-4  | InstructBLIP |
> | ------------ | ------------ | ---------- | ------------ |
> | Vanilla      | 1491.56      | 731.87     | 1271.54      |
> | VTI [1]      | 1495.47      | 728.42     | 1275.43      |
> | VASparse [2] | 1516.43      | 757.24     | 1281.32      |
> | CMI-VLD [3]  | 1504.35      | 742.13     | 1277.13      |
> | **ARTS**     | **1520.68**  | **764.38** | **1334.76**  |
>
> 2. We additionally include **OPERA** and **SID** as baselines in our **LLaVA-1.5-13B** experiments for a more comprehensive comparison. It can be seen that **ARTS outperforms both** of them.
>
> | Method   | LLaVA-1.5-13B  |
> | -------- | ----------- |
> | Vanilla  | 1504.15    |
> | OPERA     | 1502.17     |
> | SID     | 1512.43     |
> | **ARTS** | **1517.22** |
>
>
> ---
>
> ## **W2: Missing evaluation of ARTS on recent LVLM models**
>
> We include additional experiments on **LLaVA-Next** using the **MME Perception benchmark**. Unlike other methods, which often degrade or only slightly outperform the base model, **ARTS shows clear improvement**, highlighting its effectiveness even under stronger backbone architectures.
>
> | Method   | LLaVA-Next  |
> | -------- | ----------- |
> | Vanilla  | 1519.30     |
> | VCD      | 1495.47     |
> | DOLA     | 1516.43     |
> | DAMO     | 1504.35     |
> | SPIN     | 1506.37     |
> | CMI-VLD [3]  | 1522.32     |
> | VASparse [2] | 1520.98     |
> | VTI  [1]    | 1502.59     |
> | **ARTS** | **1526.68** |
>
> ---
>
> ## **W3: More hallucination-oriented benchmarks required for evaluation**
>
> 1. We add experiments on the **CHAIRs [4]** using **LLaVA-1.5-7B**, with results shown below:
>
> | Method   | CHAIR-S  | CHAIR-I  |
> | -------- | -------- | -------- |
> | Vanilla  | 46.2     | 12.0     |
> | DOLA     | 45.3     | 11.7     |
> | VCD      | 46.9     | 13.2     |
> | SID      | 44.7     | 11.2     |
> | OPERA    | 33.0     | 10.9     |
> | DAMO     | 34.6     | 11.3     |
> | **ARTS** | **31.8** | **10.5** |
>
> 2. We add experiments on the **MMBench [5]** using **LLaVA-1.5-7B**, with results shown below:
>
> | Method   | Accuracy  |
> | -------- | --------- |
> | Vanilla  | 64.57     |
> | VCD      | 64.13     |
> | OPERA    | 64.32     |
> | SID      | 64.28     |
> | CMI-VLD  | 64.77     |
> | VASparse | 64.62     |
> | VTI      | 64.41     |
> | **ARTS** | **64.79** |
>
> ---

---

> ### Author Response · Authors · 2025-11-24
>
> ## **W4: Hyperparameters (e.g., starting layer K, retention ratio P) are sensitive**
>
> 1. We thank the reviewer for pointing this out. While ARTS introduces two hyperparameters—**starting layer K** and **retention ratio P**—we find that both are easy to select and do not require extensive tuning. For **K**, pruning is most effective when applied to **middle layers**, making the selection range narrow and well-constrained. For **P**, we consistently set it below **0.5** across all models and tasks, as higher values tend to retain more redundancy.
>
> 2. We appreciate the reviewer’s concern regarding hyperparameter sensitivity. However, **model-specific tuning is common practice**—not only in hallucination mitigation, but also in reasoning and training tasks. Moreover, benchmarks target **different hallucination types**: MME Perception covers 10 diverse tasks, while POPE focuses on object hallucination. Hence, it is expected that optimal K and P vary across datasets.
>
> 3. According to Section 7.2, **as long as K is from middle layers and P < 0.5**, ARTS consistently outperforms the base model—even if not fully optimized. This shows that ARTS is **stable and deployable with different hyperparameters**.
>
> ---
>
> ## **W5: Lack of trade-off between performance gain and computational overhead**
>
> While ARTS introduces extra overhead from its scoring and selection modules, the **token pruning fully compensates** for it. As shown in **appendix E**, the **overall computational cost is lower than the base model**, demonstrating that ARTS improves hallucination robustness **without increasing inference burden**.
>
> ---
>
> ## **W6: Baselines like OPERA don’t follow the original settings**
>
> 1. Here we report the experimental results of **OPERA** on defaults beam size =5.
>
> ### 1. MME perception dataset:
>
> | Method   | LLAVA-1.5-7B | MiniGPT-4  | InstructBLIP | mPLUG-Owl2  |
> | -------- | ------------ | ---------- | ------------ | ----------- |
> | Vanilla  | 1491.56      | 731.87     | 1271.54      | 1459.54     |
> | OPERA(beam size=5)   | 1509.15      | 744.48     | 1291.57      | 1466.27     |
> | **ARTS** | **1520.68**  | **764.38** | **1334.76**  | **1474.29** |
>
> ### 2. POPE dataset:
>
> | Model            | Setting | ACC   | F1    |
> | ---------------- | ------- | ----- | ----- |
> | **LLAVA-1.5-7B** | Vanilla | 79.66 | 82.39 |
> |                  | OPERA(beam size=5)     |  80.17|  82.87|
> |                  | **ARTS**    | **80.85** | **83.71** |
> | **MiniGPT-4**    | Vanilla | 66.62 | 59.06 |
> |                  | OPERA(beam size=5)     |  66.87| 60.50 |
> |                  | **ARTS**    | **67.63** | **61.23** |
>
> ---
>
> 2. For all other baselines, we **adopt the official settings from their original papers** to ensure a **fair comparison**.
>
> ---
>
> ## **W7: Missing ablation: the effect of the number of clusters**
>
> In our method, the goal is to **separate sink tokens from other tokens**, which naturally forms a **binary classification problem**. Therefore, we set the number of clusters to **2**. Therefore, there is no need to use more clusters
>
> Moreover, existing studies have shown that **sink tokens exhibit attention weights significantly higher**—often by orders of magnitude—than those of normal tokens. This large gap makes **2-means clustering highly effective** in distinguishing the two groups.
>
> ---

---

> ### Comment · Reviewer_YxSQ · 2025-11-28
>
> Thank you for your response. I am still confused about why ARTS achieves better performance on LLaVA-1.5-7B (1520.68 on MME) than on LLaVA-1.5-13B (1517.22 on MME). This observation appears to contradict the expected scaling law.

---

> > ### Author Response · Authors · 2025-11-28
> >
> > Thanks for your reply and raising this insightful question.  Here we will clarify this question.
> >
> > 1. **Actually, this observation does not contradict the scaling law; it actually reflects it.** Since LLaVA-1.5-13B has substantially more parameters and stronger representational capacity than LLaVA-1.5-7B, its ability to handle hallucinations is inherently stronger than the 7B model’s. As a result, the headroom for ARTS to further improve the 13B model is smaller compared to the 7B model. Accordingly, it is reasonable that 7B model comes out slightly above 13B model.
> >
> > 2. Additionally, although ARTS yields a marginally higher MME score on the 7B model, the overall capability of the 13B model remains stronger in general, in line with scaling-law behavior.

---

### Official Review · Reviewer_rFz3 · 2025-10-30

**Soundness:** 3
**Presentation:** 3
**Contribution:** 2
**Rating:** 4
**Confidence:** 3

**Summary:**

The paper introduces a training-free decoding strategy aimed at mitigating hallucinations in LVLMs. The authors identify a novel source of hallucination—redundant or weakly aligned visual tokens in intermediate decoder layers—that disrupt multimodal reasoning. ARTS reinforces informative visual signals by reinjecting original visual embeddings and employs sink–visual attention to evaluate token importance. By utilizing sink tokens as global information anchors, ARTS effectively prunes redundant visual tokens at intermediate layers without additional training. Extensive experiments across multiple LVLMs and benchmarks demonstrate consistent improvements in factual accuracy and lower hallucination rates, highlighting ARTS’s robustness and generalizability.

**Strengths:**

1. The paper reframes the hallucination problem by highlighting redundant token accumulation as a new perspective, extending beyond conventional explanations centered on insufficient visual attention or weak alignment.

2. The use of sink–visual attention for redundancy estimation is creative and theoretically justified. Exploiting sink tokens as global semantic anchors is both efficient and intuitively interpretable.

**Weaknesses:**

1. Sink-token detection and redundancy scoring (via clustering and cross-token attention analysis) introduce additional computation during decoding, partially offsetting the benefits of token pruning, especially for large-scale inference.

2. ARTS assumes that attention correctly reflects token importance. In noisy or adversarial visual conditions, this assumption may not hold, potentially leading to incorrect pruning decisions.

3. There is a lack of comparison with current state-of-the-art (SOTA) models, and the experimental results are not satisfactory. Compared with other SOTA methods such as MCA[1], PATCH[2], and Less is More[3], the performance is not clearly superior.

[1] MCA-LLaVA: Manhattan Causal Attention for Reducing Hallucination in Large Vision-Language Models

[2] From Pixels to Tokens: Revisiting Object Hallucinations in Large Vision-Language Models

[3] Less is More: Mitigating Multimodal Hallucination from an EOS Decision Perspective

**Questions:**

Please refer to Weaknesses. If the author can respond to my question directly, I will increase my score.

---

> ### Author Response · Authors · 2025-11-24
>
> We sincerely appreciate your review and the valuable suggestions provided.
>
> ---
>
> ## **W1: Sink-token detection and redundancy scoring introduce additional computation during decoding**
>
> 1. While ARTS introduces additional computation through its auxiliary scoring and selection modules, the token pruning mechanism is designed to fully compensate for this overhead. As shown in **appendix E**, the overall computational cost of ARTS is even **lower than that of the base model**.
>
> 2. Moreover, ARTS is the **first decoding-time method** for hallucination mitigation that **reduces computation by directly pruning tokens**. In contrast, nearly all existing methods increase inference cost, either by only adding extra decoding steps. We add the comparison of memory usage across methods in appendix E (Table 9).
>
>
> ## **W2: In noisy or adversarial visual conditions, visual-sink attention may potentially lead to incorrect pruning decisions**
>
> Thank you for the suggestion. Our current study focuses on **benign conditions** and has been validated across multiple benchmarks and LVLM architectures. **Extending ARTS to noisy or adversarial conditions is beyond the present scope**; we view robustness as a complementary direction and will pursue it as future work with dedicated threat models, defenses, and evaluations. (Note that POPE’s “Adversarial” setting refers to clean MSCOCO images paired with intentionally challenging questions designed to induce object hallucination, which is different “adversarial” or “noisy” in the broader literature that visual conditions denote perturbed, corrupted, or adversarially attacked images that degrade the visual signal itself.)
>
> ---
>
> ## **W3: Lack of comparison with current state-of-the-art models**
>
> We appreciate the reviewer’s suggestion regarding comparisons with recent **SOTA methods** such as **MCA [1]**, **PATCH [2]**, and **Less is More [3]**. However, it is important to highlight a key distinction: **ARTS is a training-free, decoding-time method** for hallucination mitigation, whereas [1], [2], and [3] all require **additional post-training on large-scale datasets**.
>
> While these methods may achieve higher absolute performance, they require significant computation and data for post-training. In contrast, **ARTS offers consistent improvements across multiple LVLMs and benchmarks without any extra training or model changes**, making it a **lighter and more practical solution**.
>
> ---

---

> > ### Comment · Reviewer_rFz3 · 2025-11-25
> >
> > Thank you for the reviewer’s response. I am not satisfied with the answer to W2. My view is that when the user’s query is unrelated to the image (which is a very common occurrence), ARTS can still cause confusion. Therefore, I will maintain my score.

---

> ### Author Response · Authors · 2025-11-25
>
> We are glad that we resolved most of your concerns and would like to provide additional clarification on this remaining point!
>
> We apologize for initially misunderstanding your comment as referring to adversarial attacks in the sense of intentional security threats. While such attacks are out of the current scope, our paper does evaluate under adversarial or noisy visual conditions—specifically via the VizWiz dataset [4], which is included in our benchmark.
>
> In VizWiz, both images and questions are collected by blind users in real-world settings. As a result, many images are blurry, low-quality, or misaligned, and a significant portion of the questions are ambiguous or unrelated to the visual content. For example, a randomly captured photo of a computer screen may be paired with the question "Who is this mail for?", to which the correct answer should be "Unanswerable". Further dataset details can be found in [4].
>
> **Our results on VizWiz (Table 4) support the claim that ARTS is robust under such noisy or adversarial-like scenarios.**
>
> While other methods exhibit clear performance degradation on VizWiz due to its noisy and ambiguous nature, ARTS remains effective and robust. This suggests that ARTS can still **correctly reflect token importance**, even when the visual input is noisy or weakly relevant. By identifying and **discarding irrelevant or misleading visual tokens**, ARTS effectively reduces the impact of noisy signals. In contrast, other methods focus solely on enhancing or aggregating visual information may inadvertently amplify noise under such conditions, leading to performance degradation.
>
>
> [4] VizWiz Grand Challenge: Answering Visual Questions from Blind People

---

> > ### Comment · Reviewer_rFz3 · 2025-11-26
> >
> > Thank you for your response. I will maintain my score.

---

### Official Review · Reviewer_VoWh · 2025-10-31

**Soundness:** 3
**Presentation:** 3
**Contribution:** 3
**Rating:** 6
**Confidence:** 3

**Summary:**

ARTS is a training‑free, mid‑layer decoding strategy for LVLMs. It detects sink tokens (via 2‑means on column‑wise attention), reinjects original visual embeddings to refresh visual signal, and then keeps only top‑p visual tokens by sink‑visual attention, pruning the rest. This aims to remove redundant/noisy mid‑layer visual tokens that contribute to hallucinations. Across MME/POPE/VizWiz/MM‑Vet and multiple backbones (7B and 13B), ARTS consistently improves factual alignment over strong decoding baselines.

**Strengths:**

1. Fresh diagnosis & evidence for mid‑layer redundancy. Section 3.1 explains infomation flow rationale.

2. Simple, training‑free, model‑agnostic insertion point. ARTS operates at inference in intermediate layers (no finetuning.

3. The use of sink tokens as anchors is novel. Prior works reallocate away from sinks; ARTS leverages sinks to rank visual tokens.

4. Consistent gains across multi‑backbone, multi‑benchmark

**Weaknesses:**

1. There’s no runtime or memory comparison. This is important for deployment to compare how much time and compute needed for extra steps.

2. No direct “random vs ARTS” at the same K/P on the same tasks. Fig. 1b shows random pruning helps in mid layers; Tables 2–4/5/7 show ARTS improvements, but a side‑by‑side matched comparison is missing.

3. Sink detection stability unreported. In section 5.1, it proposes 2‑means on column‑wise sums but provides no statistics (e.g., typical number of sink tokens, layerwise stability, per‑head variability). Only Table 1 indirectly shows the importance of sinks.

**Questions:**

1. What are the latency/throughput/memory effects of ARTS vs. vanilla and vs. pruning baselines (visual‑text/CLS)? Please report wall‑clock, FLOPs (pre‑/post‑pruning), and GPU memory

2. Can you add a matched‑budget comparison (same K and P) against random pruning on MME perception and POPE? Fig. 1b suggests random helps mid‑layer; quantifying ARTS’s lift over random would isolate its value.

3. Please report typical number of sinks per layer/input, variance across heads, and stability across images/prompts; Table 1 (p. 5) shows sinks matter, but not how many you find.

---

> ### Author Response · Authors · 2025-11-24
>
> We sincerely appreciate your review and the valuable suggestions provided.
>
> ---
>
> ## **W1 & Q1: No runtime or memory comparison**
>
> 1. We add a comprehensive computational efficiency comparison between different pruning methods (ARTS vs. vanilla and vs. visual‑text/CLS) with different retain ratios **in appendix E**  (Table 10). ARTS incurs only marginal increases in FLOPs, latency, and other computational cost.
>
> 2. We provide a comprehensive computational efficiency comparison of different hallucination-alleviation methods in **appendix E  (Table 9)**. ARTS is the only method that consumes less GPU memory than the base model.
>
> ---
>
> ## **W2 & Q2: No direct “random vs ARTS” at the same K/P on the same tasks**
>
> We add comparison experiments between random pruning and ARTS on MME perception and POPE datasets.
>
> ### 1. MME perception dataset:
>
> | Method   | LLAVA-1.5-7B | MiniGPT-4  | InstructBLIP | mPLUG-Owl2  |
> | -------- | ------------ | ---------- | ------------ | ----------- |
> | Vanilla  | 1491.56      | 731.87     | 1271.54      | 1459.54     |
> | Random   | 1497.39      | 735.12     | 1282.26      | 1462.45     |
> | **ARTS** | **1520.68**  | **764.38** | **1334.76**  | **1474.29** |
>
> ### 2. POPE dataset:
>
> | Model            | Setting | ACC   | F1    |
> | ---------------- | ------- | ----- | ----- |
> | **LLAVA-1.5-7B** | Vanilla | 79.66 | 82.39 |
> |                  | Random    |  80.21|  82.98|
> |                  | **ARTS**    | **80.85** | **83.71** |
> | **MiniGPT-4**    | Vanilla | 66.62 | 59.06 |
> |                  | Random    |  67.05| 60.22 |
> |                  | **ARTS**    | **67.63** | **61.23** |
>
> ---
>
> ## **W3 & Q3: Sink Token Analysis**
>
> The sink token phenomenon refers to the behavior during LVLM decoding where attention weights disproportionately concentrate on a small number of specific tokens, regardless of input content. Here, we report the number of sink tokens identified by our 2-means clustering algorithm. We observe that sink tokens first appear at the 3rd decoder layer, and their number remains consistent across all deeper layers.
>
> Our method locates sink tokens **at a single mid-level decoder layer and uses the average attention across all heads**. The stability across different layers and attention heads does not affect our method. This analysis is conducted purely for transparency.
>
> | Layer | Num of Sink Token | Layer | Num of Sink Token | Layer | Num of Sink Token |
> | ----- | ----------------- | ----- | ----------------- | ----- | ----------------- |
> | 1     | 36                | 13    | 2                 | 25    | 2                 |
> | 2     | 31                | 14    | 2                 | 26    | 2                 |
> | 3     | 2                 | 15    | 2                 | 27    | 2                 |
> | 4     | 2                 | 16    | 2                 | 28    | 2                 |
> | 5     | 2                 | 17    | 2                 | 29    | 2                 |
> | 6     | 2                 | 18    | 2                 | 30    | 2                 |
> | 7     | 2                 | 19    | 2                 | 31    | 2                 |
> | 8     | 2                 | 20    | 2                 | 32    | 2                 |
> | 9     | 2                 | 21    | 2                 |       |                   |
> | 10    | 2                 | 22    | 2                 |       |                   |
> | 11    | 2                 | 23    | 2                 |       |                   |
> | 12    | 2                 | 24    | 2                 |       |                   |
>
> ---

---

> > ### Comment · Reviewer_VoWh · 2025-11-27
> >
> > Thanks for the response. Most of my concerns are addressed. I'll keep my rating and tend to accept.

---

> > > ### Author Response · Authors · 2025-11-28
> > >
> > > Thanks for your reply. We are glad that we resolved your concerns, and we sincerely appreciate your recognition of our work.

---

### Official Review · Reviewer_VkP5 · 2025-11-01

**Soundness:** 1
**Presentation:** 1
**Contribution:** 1
**Rating:** 2
**Confidence:** 4

**Summary:**

The paper introduces ARTS, a training-free decoding strategy to reduce hallucinations in LVLMs. Unlike prior work that attributes hallucinations mainly to insufficient attention to relevant information, this paper identifies that redundant or noisy visual tokens in intermediate decoder layers that mislead reasoning. ARTS first re-injects original visual embeddings to preserve essential visual information and then employs a sink-token-based cross-attention mechanism to assess token importance, pruning redundant visual tokens dynamically. Experiments across multiple LVLMs (LLaVA-1.5, mPLUG-Owl2, InstructBLIP, MiniGPT-4) and datasets (MME, POPE, VizWiz, MM-Vet) show that ARTS outperforms existing training-free methods (e.g., DoLa, VCD, DAMO, SID, OPERA) in reducing hallucinations while maintaining reasoning capability.

**Strengths:**

- **Clear Writing.** The paper's writing is clear and easy to follow.
- **Novel Perspective.** The paper identifies visual token redundancy in intermediate layers as a new and underexplored source of hallucinations, extending beyond the common “insufficient attention” explanation.
- **Simple method.** ARTS is a training-free and lightweight decoding approach, making it practical to integrate into existing LVLMs without retraining or architecture changes.

**Weaknesses:**

+ **Incremental improvement of ARTS.** The biggest concern I have is the (very) limited performance gains. Compared to the baselines, ARTS seems to be not strong. For example, in Table 3, most of the improvement is within 1 point, without reporting statistical significance. This may raise questions about practical significance versus added inference complexity.
+ **Lack of Intuitive/Theoretical Grounding.** The design of the current method lacks intuitive explanation and theoretical grounding. For example, there is almost no explanation of the 2-means clustering and the assumption of the Euclidean distance measure; there is no discussing of the form of re-injecting the visual tokens' information back to the hidden states.
+ **Not Comprehensive Enough Evaluation Benchmarks.** It looks like the authors break down the POPE benchmark into multiple subsets to report (Table 3) instead of reporting the average scores. The paper also lacks the results on CHAIRs benchmark.
+ **Missing Crucial Baselines.** Two recently published strong baselines are not discussed in the paper:
  + Liu, Shi, Kecheng Zheng, and Wei Chen. "Paying more attention to image: A training-free method for alleviating hallucination in lvlms." European Conference on Computer Vision. Cham: Springer Nature Switzerland, 2024.
  + Li, Zhuowei, et al. "The hidden life of tokens: Reducing hallucination of large vision-language models via visual information steering." ICML, 2025.
+ **Multiple Formatting Issues.**
  + Tables' caption needs to be placed on top of the table body;
  + The citation format is inconsistent and incorrect.

**Questions:**

See above (Weaknesses).

---

> ### Author Response · Authors · 2025-11-24
>
> We sincerely appreciate your review and the valuable suggestions provided.
>
> ## **W1: Limited performance on POPE dataset & significance versus added inference complexity.**
>
> 1. It is common that various methods—including ARTS and prior baselines—show limited or even negative changes on certain subsets or settings on POPE dataset. For example, when using LLAVA-1.5-7B as the base model, **the COCO–Random subset provides very limited headroom for all methods**. That is because the diverse object distribution in COCO and the uniform sampling in the Random setting, hallucination-prone cases are less concentrated, making the task easier and limiting the potential for improvement across all methods. However, ARTS demonstrates much stronger improvements on other subsets. For instance, on the challenging Adversarial subset, where more hallucination-prone objects are likely to be sampled, ARTS surpasses the strongest baseline (DAMO) by +0.98 ACC / +1.27 F1, showing a clear reduction of hallucinations where models typically struggle the most.
>
> 2. The overall gains of ARTS remain substantial, across the full POPE dataset. To provide a comprehensive view, we report the **average accuracy and F1** of ARTS and the vanilla models (LLAVA-1.5-7B and MiniGPT-4, InstructBLIP, and mPLUG-Owl2) on the MSCOCO, A-OKVQA, and GQA datasets under all three POPE settings (Random, Popular, and Adversarial). **It is clear that ARTS can work on various LVLM architectures on object hallucination mitigation task, demonstrating that its effectiveness generalizes across architectures.**
>
> | Model            | Setting | ACC   | F1    |
> | ---------------- | ------- | ----- | ----- |
> | **LLAVA-1.5-7B** | Vanilla | 79.66 | 82.39 |
> |                  | **ARTS**    | **80.85** | **83.71** |
> | **MiniGPT-4**    | Vanilla | 66.62 | 59.06 |
> |                  | **ARTS**    | **67.63** | **61.23** |
> | **mPLUG-Owl2**   | Vanilla | 82.36 | 82.53 |
> |                  | **ARTS**    | **83.97** | **84.08** |
> | **InstructBLIP** | Vanilla | 80.41 | 82.38 |
> |                  | **ARTS**    | **82.03** | **83.24** |
>
> 4. **While the POPE dataset primarily targets object hallucination, its scope is relatively narrow.** In contrast, the MME Perception benchmark offers a more comprehensive evaluation of hallucinations across 10 diverse tasks, including fine-grained perception, attribute grounding, and spatial reasoning. On this benchmark, ARTS delivers substantial improvements, **achieving 30+ average gains across models—and over 60 points on InstructBLIP**—highlighting its strong generalization to broader hallucination scenarios.
>
>
> 3. **ARTS effectively alleviates hallucination without compromising generalization ability**, whereas nearly all other decoding-based methods designed for hallucination mitigation noticeably degrade overall model performance (e.g., on VizWiz, MM-Vet, and MME reasoning tasks). Moreover, on larger models such as LLAVA-1.5-13B, ARTS continues to deliver clear improvements in hallucination reduction, while other methods either exhibit performance drops or merely maintain their original performance.
>
> 4. Here we set **temperature to 0** for fair comparison and this removes sampling noise and allows performance differences to more faithfully reflect the intrinsic capability gaps between methods.
>
> 5. We provide a comprehensive computational efficiency comparison between different pruning strategies (Table 10) and hallucination-alleviation methods (Table 9) in appendix E. ARTS is the only method that consumes less GPU memory than the base model.
> ---
>
> ## **W2: no explanation of the 2-means clustering & the assumption of the Euclidean distance measure; & the form of re-injecting the visual tokens' information back to the hidden states.**
>
> 1. We have added the detailed formulation and optimization process of the 2-means clustering in the **appendix D**.
>
> 2. Assumption of the Euclidean distance measure: In our method, we use clustering to identify sink tokens and first compute each token’s column-wise averaged attention score. This is a 1D clustering problem, where each token is represented by a scalar attention score. In such cases, **all standard distance measures (e.g., Euclidean, Manhattan) are effectively equivalent**, as they reduce to the absolute value of pairwise differences.
>
> 3. The form of re-injecting: We simply extract the visual embeddings from the vision encoder and directly add them back to the visual portion of the hidden states before pruning.
>
> ---

---

> ### Author Response · Authors · 2025-11-24
>
> ## **W3: No reporting the average scores & lacks the results on CHAIRs benchmark.**
>
> 1. We report the average ACC and F1 scores of ARTS on for LVLM architectures on the MSCOCO, A-OKVQA, and GQA datasets under all three POPE settings in W1.
>
> 2. We add experiment results on CHAIRs benchmark on **LLAVA-1.5-7B**:
>
> | Method  | CHAIR-S | CHAIR-I |
> | ------- | ------- | ------- |
> | Vanilla | 46.2    | 12.0    |
> | DOLA    | 45.3    | 11.7    |
> | VCD     | 46.9    | 13.2    |
> | SID     | 44.7    | 11.2    |
> | OPERA   | 33.0    | 10.9    |
> | DAMO    | 34.6    | 11.3    |
> | **ARTS**    | **31.6**    | **10.5**    |
>
> ---
>
> ## **W4: Missing baselines PAI[1], VISITA[2]**
>
> 1. PAI [1] observes that during decoding the model’s attention concentrates on certain fixed tokens (sink tokens) while image tokens receive very low attention scores; thus PAI assigns larger attention weights to image tokens. However, we find that during decoding much of the image‐token information has already flowed into the sink tokens, so simply increasing the model’s focus on image tokens is insufficient because the bulk of image‐token information is already redirected to other tokens.
>
> 2. VISITA [2] discovers that during decoding there is visual‑information loss leading to semantic inconsistency; while VISITA focuses on supplementing visual information, it does not eliminate redundant information — by contrast, our ARTS both enhances visual information and removes redundant information.
>
> 2. Here we add experiments using 4 base models on MME perception benchmark. It can be seen that our method has stronger performance.
>
> | Method   | LLAVA-1.5-7B | MiniGPT-4  | InstructBLIP | mPLUG-Owl2  |
> | -------- | ------------ | ---------- | ------------ | ----------- |
> | Vanilla  | 1491.56      | 731.87     | 1271.54      | 1459.54     |
> | PAI [1]     | 1487.31      | 733.67     | 1257.38      | 1461.27     |
> | VISITA [2]  | 1503.12      | 756.24     | 1280.42      | 1471.28     |
> | **ARTS** | **1520.68**  | **764.38** | **1334.76**  | **1474.29** |
>
> ---
>
> ## **W4: Incorrect table format & citation format.**
>
> Thanks for pointing this out. We have revised the table format and citation format in our new version.
>
> ---

---

### Meta-Review · Area_Chair_mGTB · 2026-01-06

**Summary:**

Key concerns include:
1. lack of theoretical/intuitive grounding
2. incomplete baselines
3. no runtime/memory comparison or random pruning vs. ARTS contrast
4. hyperparameter sensitivity

 Authors addressed these via supplementary experiments  and explanations.

**Reviewer Concerns:**

The following ones are outstanding:

Reviewer VkP5:  Incremental performance gains (reviewer did not respond to rebuttal; authors addressed dataset headroom but did not report statistical significance).

Reviewer rFz3:  Robustness to unrelated querie

**Reviewer Scores:**

Reviewer VkP5 may consider the feedback from authors such as added POPE average scores,  added CHAIRs benchmark, etc.

---

### Decision · Program_Chairs · 2026-01-26

Reject